# Development and Field Validation of Wireless Sensors for Railway Bridge Modal Identification

Federico Zanelli *, Nicola Debattisti, Marco Mauri, Antonio Argentino and Marco Belloli *

Department of Mechanical Engineering, Politecnico di Milano, 20156 Milan, Italy;
nicola.debattisti@polimi.it (N.D.); marco.mauri@polimi.it (M.M.); antonio.argentino@polimi.it (A.A.)
* Correspondence: federico.zanelli@polimi.it (F.Z.); marco.belloli@polimi.it (M.B.); Tel.: +39-02-2399-8377 (F.Z.)

**Abstract:** Bridges are strategic infrastructures which are subject to degradation during their lifetime. Therefore, structural health monitoring is becoming an essential tool in this field to drive maintenance activities. Conventional vibration monitoring systems relying on wired sensors present several limitations for continuous monitoring projects on a huge number of structures. In this work, a smart wireless monitoring system is developed for bridge modal identification with the aim of providing an alternative tool to wired sensors in this field. The main peculiarities of the designed wireless accelerometers are the low cost, the ease of installation on the structure, and the long-term autonomy granted by the use of energy harvesting techniques. To assess their measurement performance, some prototypes were installed for a field test on a railway bridge and significant data were acquired. Through the processing of the collected data, bridge main natural frequencies were estimated, and their values were in good agreement with the reference ones obtained with a conventional system. The assessment of the developed solution paves the way to the instrumentation of many bridges with the aim of performing continuous monitoring activities using simple diagnostic indicators, such as the variation of frequencies in time.

**Keywords:** structural health monitoring; bridges; predictive maintenance; wireless sensors; MEMS accelerometers; vibrations





## 1. Introduction

The economic growth and development of a country depends significantly on the lifetime performance of its civil infrastructures. Among them, bridges represent one of the most vulnerable and crucial infrastructural connection nodes [1]. The degradation in time of this kind of infrastructures is due to several factors, among which aggressive environment, increasing loads due to higher traffic volumes, and lack of maintenance are reported [2]. For these reasons, the monitoring of bridges and viaducts is gaining more and more importance nowadays [3].

Structural health monitoring (SHM) focuses on controlling in time the loading conditions the structure is subject to, to assess its performance and to detect in advance damage and deterioration with the aim of performing predictive maintenance activities [4]. Recently, innovative approaches based on artificial intelligence and neural networks have been proposed for damage identification in complex structures [5,6]. A more traditional SHM branch deals with vibration-based methods, which allow the identification of structural damages by detecting a change in the bridge dynamic behavior [7]. The key physical parameter to be measured in this case is acceleration.

In recent years, the research in the field of monitoring activities on bridges has become more intense, due to the need for obtaining indications on the structural health to drive the maintenance operations and avoid failures with disastrous consequences [8].

The standard approach to carry out those activities is to instrument the bridge with monitoring systems composed of wired vibrations sensors (such as accelerometers or

velocimeters) positioned in suitable points of the structure with the aim of measuring the structural natural frequencies, reconstructing mode shapes, and detecting the damping value [9]. The variation in time of these two parameters represents a basic method to perform SHM of a civil infrastructure. Since it is important to observe the trend of these parameters during the life of the structure, the main goal is continuous monitoring in time [10].

The use of continuous monitoring systems for maintenance purposes is already established in many engineering fields, such as on railway vehicles [11] and wind turbines [12]. Nonetheless, in the case of viaducts and bridges, continuous monitoring is not an easy task to perform with the currently available instrumentation. Established monitoring systems are in fact characterized by the presence of wired sensors that present several limitations in this sense.

Firstly, the permanent installation of wired sensors on railway bridges is not always feasible due to their structural conformation, and it can result very expensive in terms of both time and hardware cost. In most cases, the deck must be suitably equipped with ducts and other fittings devoted to the protection of communication cables. The installation time tends to be very long, and this issue can be even worsened by the fact that railway bridges are operative infrastructures which rarely can be closed for long time windows reserved to monitoring system installations.

Secondly, the maintenance over a long time of the monitoring system can be costly, both from the hardware and the software point of view. Signal cables are subject to a constant degradation in time which require maintenance activities on them. Moreover, if a transducer inside the monitoring network presents some issues, it is usually difficult to find the root cause and fix the problem in the wired network.

Thirdly, the large quantity of data collected by these systems requires a complex IT infrastructure to be managed in time. Big data analyses are often required to process data and obtain significant diagnostic information, resulting in an additional complexity to the problem.

For these reasons, wireless sensors can represent a suitable alternative to wired ones when analogous measurement performances are achieved. The recent technological advances in the electronic field have made it possible to create increasingly miniaturized and high-performance components, reducing production costs of innovative sensors [13]. Wireless sensors have been developed for monitoring activities in various complex fields, such as monitoring of wind-induced vibration on transmission line conductors [14], dynamics of freight wagon vehicle and the infrastructure [15,16], and historical buildings [17]. Some applications of wireless sensors can also be found in the active vibration control field [18,19].

The main features of sensor nodes to be suitably employed for bridges' SHM are the measurement performances, the easiness of installation, and the long autonomy to grant a permanent installation on the structure.

The performance of MEMS accelerometers to be mounted on printed circuit boards (PCB) available in the market nowadays is closer to the ones of wired models [20,21]. Great improvements have been especially made in terms of noise density, which is an aspect of crucial importance for SHM of civil structures [22,23]. Moreover, some models of these low-cost accelerometers are characterized by low power consumption.

To be mounted easily on the structure, sensor nodes must rely on wireless communication for the data transmission. Among the several available protocols (Zigbee, LoRa, Wi-Fi, etc.), Bluetooth low energy (BLE) is drawing interest for monitoring activities thanks to the a good trade-off between data rate, low consumption, and communication range [24,25].

In the end, sensor nodes must guarantee a long autonomy. This can be achieved by choosing very low power components to realize the electronic board and adopting a smart acquisition logic to maintain the sensor in a low-power condition (usually called "sleep" mode) when data gathering is not required. On the other hand, an inflow of energy must be guaranteed to recharge the sensor battery using an energy harvester [26]. Different types

of harvesting sources are described in the literature, such as piezoelectric harvesters [27], micro wind turbines [28] and thermo-electric generators (TEG) [29]. Anyway, the best solution for structures located in an outdoor environment is photovoltaic (PV) panels, since they allow harvesting solar energy, which is a reliable source characterized by a high power density [30].

Some examples of wireless sensors devoted to SHM purposes are described in [31]; anyway, they were devices with low computational power, with a short communication range and without energy harvesting sources able to provide them with a long-term autonomy. In a more recent work [32], a comparison between different wireless sensors [32–34] is reported and hereafter reproposed in Table 1.

**Table 1.** Comparison between different wireless sensors available in literature.

| Feature | Present Paper | Xnode [30] | iMote2 [31] | Microstrain [32] |
|---|---|---|---|---|
| Sampling Rate | 8 kHz | 16 kHz | 100 kHz | 1 kHz |
| ADC Resolution (bits) | 16 | 24 | - | 20 |
| Accelerometer Noise Density | 70 µg/$\sqrt{\text{Hz}}$ | 50 µg/$\sqrt{\text{Hz}}$ | - | 25 µg/$\sqrt{\text{Hz}}$ |
| Communication protocol | BLE | 2.4 GHz radio | 802.15.4 radio | LXRS® |
| Communication range | >200 m | 1 km | 300 m | 800 m (typical) |
| On-board processing | yes | yes | yes | no |
| Energy Harvesting | yes | no | no | no |

As can be noticed, the sensor proposed in this paper is the only one relying on an energy harvesting technique to guarantee a long-term monitoring capability.

In this work, the design and development of an innovative wireless system for the continuous vibration monitoring of viaducts and bridges is described. The main advantages of the proposed system are the fact that the realized wireless accelerometers are low cost, compact, and easy to install in the point of interest on the structure. These sensors are autonomous from an energetic point of view, thanks to the sensor's low-power consumption and the use of a mini-PV panel to recharge the lithium-polymer (Li-Po) battery when an energy excess is produced [35]. This way, the realized system can be permanently mounted on the structure during its entire service life and the wireless communication helps avoid the use of any cables between sensors and the acquisition part. Moreover, the developed smart sensors are able to acquire acceleration data and perform data pre-processing activities on board, sending only significant information to the control unit to be analyzed remotely. The developed system can, therefore, represent a new tool in the field to be used for bridge modal identification purposes.

The paper is organized as follows. In Section 2, the smart wireless monitoring system developed for bridges' SHM is described. The focus is on the design of the wireless sensors equipped with MEMS accelerometers. which are the sensing nodes to be positioned on the structure. A field test arranged to evaluate the performances of the developed system is presented in Section 3; contextually, a comparison is made with the results coming from a traditional monitoring system including wired velocimeters, already installed for other activities. This comparison allowed for the validation of the proposed solution. Some preliminary results of the experimental activity are then shown in Section 4, and a basic diagnostic indicator is obtained for the case study. In the conclusions, the potential of the proposed system to be used for SHM of bridges is highlighted as well as actual limitations and future improvements are discussed.

## 2. Description of the Wireless Monitoring System

The proposed monitoring system is made up of a gateway and wireless sensor nodes, which are mounted on the structure. The sensor nodes acquire vibration data and transmit them wirelessly to the gateway, which has the role of a central acquisition unit (Figure 1). The communication protocol chosen for the wireless side is the BLE, as already pointed out in Section 1.

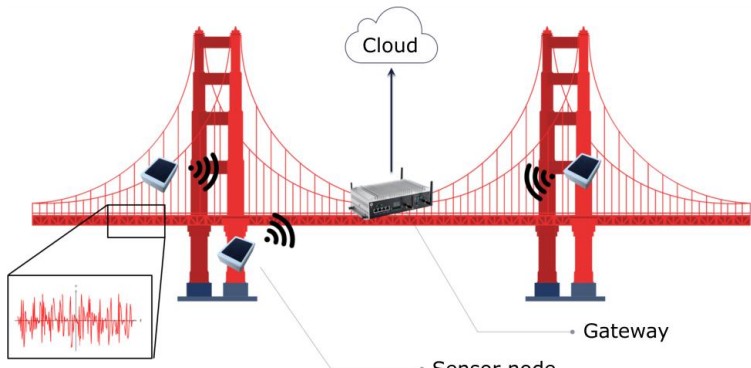

**Figure 1.** Schematization of the monitoring system to be mounted on the structure.

### 2.1. Gateway

The gateway is essentially a cabinet hosting a DC–DC converter, a Raspberry Pi4, a master communication board, and a 4G modem. The acquisition script runs on the Pi4, which is connected through serial communication to the master board. The master board is endowed with a BLE transceiver and an omni-directional external antenna to communicate with sensor nodes. Acceleration data are sent by the sensor nodes and received through the master board, which is read by the Pi4 to gather all the data and store them locally. Collected data are automatically uploaded to the cloud so that can be accessed remotely to be analyzed. Moreover, the BLE communication also allows sending of messages from the master board to the sensors. This functionality is used for changing some acquisition parameters of the sensor nodes. In fact, the gateway can wake up sensor nodes on request and send them a specific message, which can modify some configuration parameters.

The gateway is fed with a 24 VDC power supply. Due to the low power consumption of the different components, it can also be fed by a battery recharged through a photovoltaic panel suitably positioned on the side of the deck to avoid the presence of any power supply cable.

### 2.2. Wireless Sensor Nodes

The sensor nodes described in this work represent an upgrade of the ones described in [36,37]. The core of the device is the PCB, which is shown in Figure 2. The designed circuit has the function of performing measurements, processing the acquired data, and sending the results to the gateway.

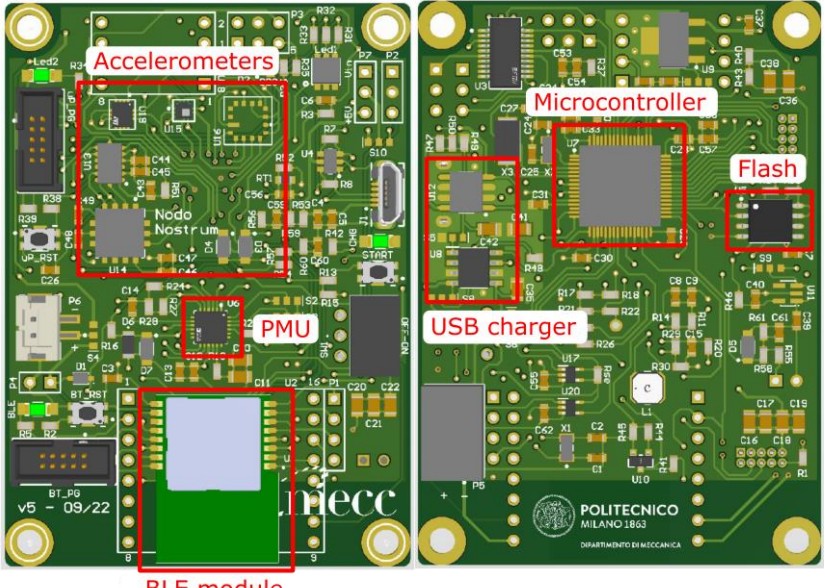

**Figure 2.** Layout of the PCB and focus on the main components.

### 2.2.1. Architecture of the Sensor Node

The main components can be divided into three main areas, depending on the task for which they were designed, which are power, logic, and communication. The power supply management is very important to guarantee a long autonomy, and it represents one of the most innovative features of the developed sensor.

The power supply of the sensor node is provided by a 3.7 V lithium polymer (Li-Po) battery and a photovoltaic (PV) panel, which allows recharging of the battery when a sufficient solar radiation is available. The component which manages these two power sources is the ADP5091, an ultralow power energy harvester power management unit (PMU). The maximum power point tracking (MPPT) control keeps the input voltage ripple in a fixed range, to obtain a DC–DC boost conversion as stable as possible. The dynamic sensing mode and no sensing mode allow the extraction of the highest possible energy from the power source. Such a component also provides the stabilized 3.3 V power supply for all components on the PCB. Moreover, a standard micro-USB port is available, which is used to fast-charge the battery and to get information about the node, like the configuration parameters stored in the flash memory.

The microcontroller is the main element of any digital device, since it is the programmable component that manages the entities of the whole system. An ultralow power Arm Cortex M4 STM32 device is used. The Cortex-M4 core features a floating-point unit (FPU) single precision, which supports all arm single-precision data-processing instructions and data types. It also implements a full set of DSP instructions and a memory protection unit (MPU). The sensing part of the board presents four MEMS accelerometers with $\pm 8$ g or $\pm 16$ g measurement range, which can communicate with the microcontroller through both the SPI and the I2C protocols. The sensor used in this work is provided by TDK InvenSense (IIM-42351), a three-axis accelerometer with low noise and low power features, a $\pm 16$ g measurement range, and a sampling frequency up to 8 kHz. The main features of the adopted MEMS accelerometer are reported in Table 2. Connected to the microcontroller, there is also a flash memory able to store custom configuration setups, lookup tables, and manage a circular buffer with the processed data. The presence of the microcontroller allows performing on-board processing on the acquired data such as the fast Fourier transform (FFT) or RMS.

**Table 2.** MEMS accelerometer (TDK IIM-42351) main features.

| Parameter | Value |
| --- | --- |
| Full-Scale Range | $\pm 2$ g, $\pm 4$ g, $\pm 8$ g and $\pm 16$ g |
| Sensitivity | 16,384 LSB/g (for $\pm 16$ g range) |
| Noise Density | 70 $\mu$g/$\sqrt{\text{Hz}}$ |
| Output Data Rate | up to 8 kHz |
| Supply Voltage | $1.8 \div 3.6$ V |
| Supply Current | 0.3 mA |
| Operating Temperature Range | $-40 \div 105$ °C |

The communication between the sensor node and the gateway takes place through a wireless communication channel, which is based on the Fanstel Bluetooth Low Energy 5.0 (Fanstel Corp., Scottsdale, AZ, USA) (BLE) stack. Such a protocol ensures a good communication range (up to 200 m in line of sight) with a low power consumption (10–500 mW).

### 2.2.2. Acquisition Procedure

The proposed monitoring system based on wireless sensor nodes implements a power saving logic, which ensures a long battery life. The goal is to provide the sensor with the longest possible autonomy, the only problem being battery ageing [38]. The power available on the photovoltaic panel is limited, and it is not constant over time, since it depends on many variables (day/night, environmental conditions, irradiation, season, dirt). For this reason, the power consumption of the node must be as low as possible.

To do so, the sensor node always works in a low-power condition in which the clock frequency is reduced, and all functions are disabled, except for the wake-on-threshold feature of the accelerometer and the BLE pairing with the gateway. The first function represents the automatic trigger for the acquisition of the acceleration data. In this condition, the accelerometer is not communicating with the microcontroller via the SPI protocol. Once the acceleration module exceeds a given threshold parameter, a logic signal (detected by an interrupt pin) is sent to the CPU to start the acquisition. The second function, instead, is kept active because the sensor node can be woken up by the gateway anytime, both to trigger an acquisition without exceeding the acceleration threshold and to change the configuration parameters stored into the flash memory (as mentioned in Section 2.1). The parameters that can be modified are the sampling frequency, the measurement range, the wake-up acceleration threshold, the acquisition period, the number of axes to sample, and the axis that triggers the wake-up condition. The useful aspect of this approach is that such parameters can be modified remotely by directly connecting to the gateway, without downloading a new firmware on each sensor node.

So, when a significant disturbance force excites the whole structure, like the one generated when a train crosses the bridge, the accelerometer triggers the node as described before. Then, the sensor node switches to the full-operating condition and samples the acceleration values with a sampling frequency equal to 200 Hz for 10 s, both on the vertical and the lateral axis. When the acquisition is over, the data are divided into some packages and then sent to the gateway, together with some other information (such as the timespan, the battery voltage, the voltage on the PV panel, the power consumption, and the temperature). Once all the data have been properly transmitted, the sensor node switches again to the low-power condition and the gateway uploads the collected data on the cloud, as shown in Figure 3.

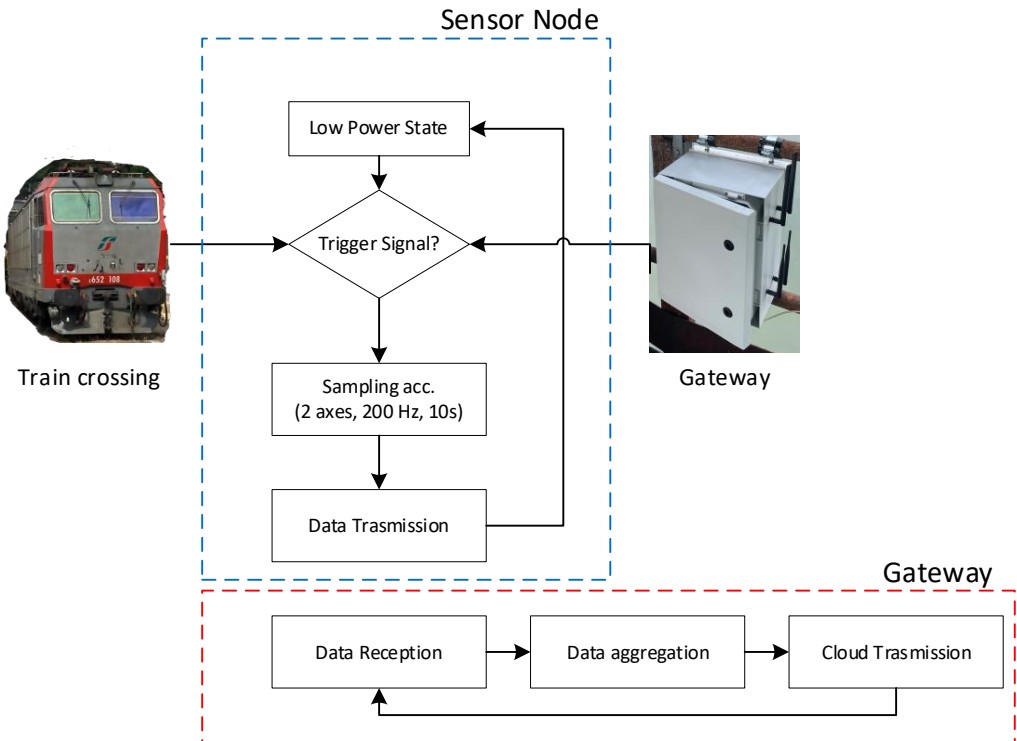

**Figure 3.** Schematization of the acquisition procedure.

## 3. Field Test on the Bressana Railway Bridge

An experimental campaign was arranged to test the developed system in the field. The purpose of this test was to verify the efficiency of the energy harvesting system equipping

the sensors, to check the wireless communication in the real monitoring context, and to understand the significance of the vibration data collected.

Moreover, a conventional monitoring system based on wired velocimeters already mounted on the bridge was used to perform a benchmark between the results obtained from the two different solutions.

The Bressana railway bridge on the Po River is located near the city of Pavia, in Lombardy. The bridge crosses the river on three spans approximately 77 m long each (Figure 4a). The bridge is constituted of a truss steel structure with a double deck; the lower deck is devoted to the railway traffic while the upper one is reserved for the vehicular traffic (Figure 4b).

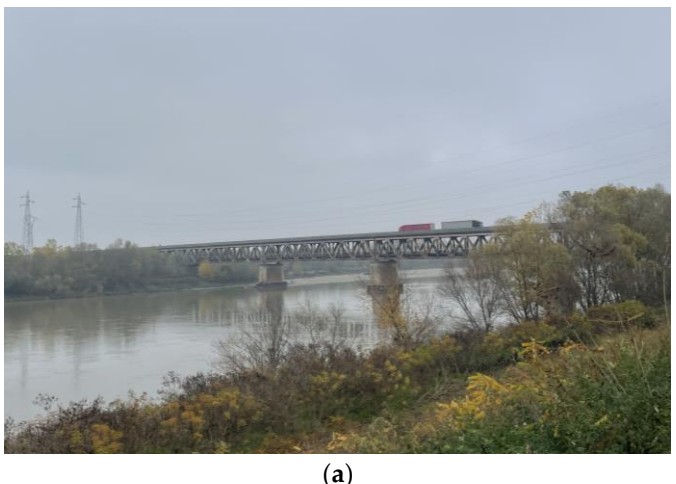 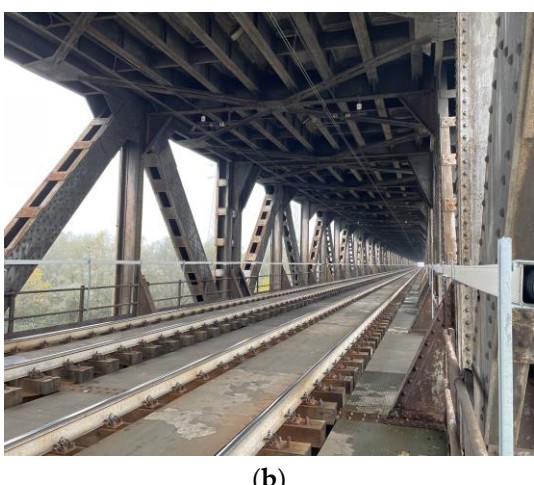

(**a**)　(**b**)

**Figure 4.** The Bressana bridge on the Po river: (**a**) Lateral view of the bridge crossing the river with heavy vehicles running on the upper deck; (**b**) internal view of the lower deck reserved to railway traffic.

First, the permanent monitoring system, already installed on the bridge for monitoring activities, is briefly described. The schematic setup of the system is shown in Figure 5. This system consists of 15 wired velocimeters, 5 for each span. The upstream velocimeters (placed at the bottom of the top view in Figure 5) are biaxial sensors that measure the lateral and the vertical velocity of the bridge in the application points. The downstream velocimeters are monoaxial sensors that measure the vertical velocity only. In this way, the natural frequencies related to the vertical and lateral modes can be estimated, as will be shown in Section 4.

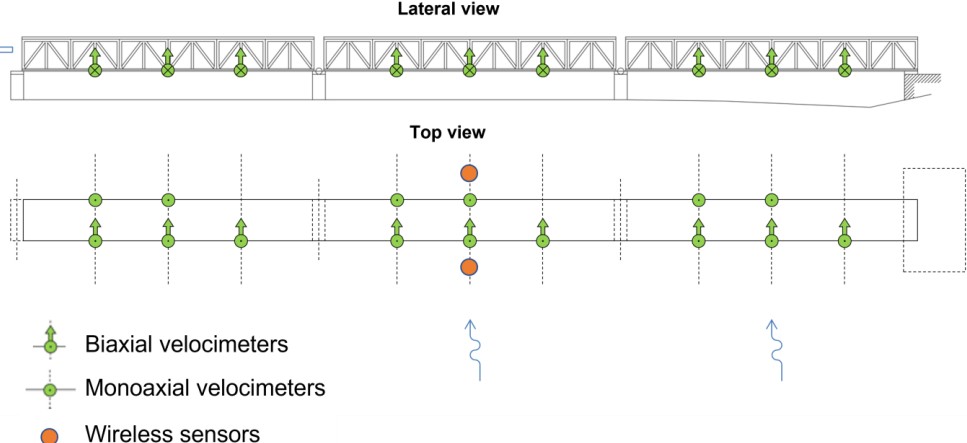

**Figure 5.** Schematic set-up of the wired velocimeters forming the permanent monitoring system.

Concerning the developed wireless monitoring system, it was easily installed on the bridge. Two sensor nodes were mounted on the pillars in the middle of the central span by using a suitable structural adhesive (Figure 6a). The gateway was instead placed in a balcony of the railway deck (Figure 6b).

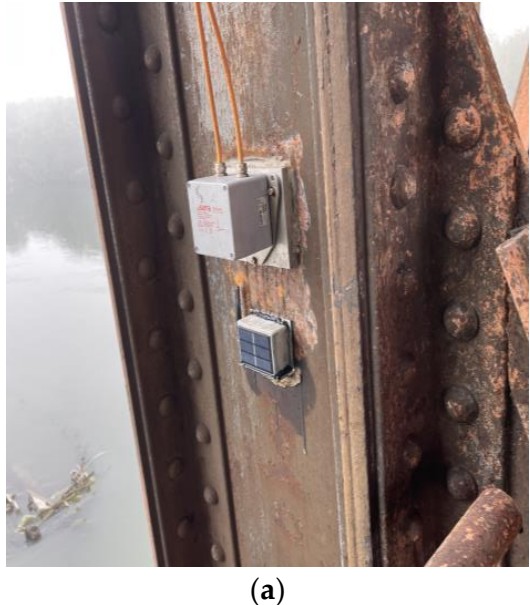

(**a**)

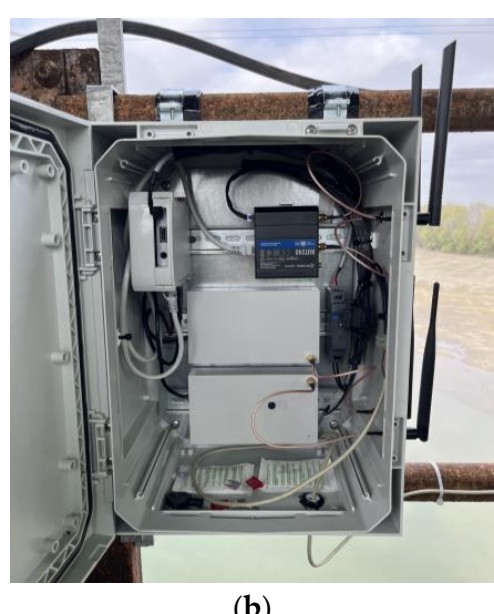

(**b**)

**Figure 6.** Wireless monitoring system mounted on the bridge: (**a**) Wireless sensor positioned in correspondence of a wired velocimeter; (**b**) gateway installed on a balcony of the railway deck.

The main features of the two different monitoring devices installed on the bridge are reported in Table 3. As can be noted, the adoption of wireless sensors for this kind of a monitoring campaign presents several advantages in case the performance of the two systems is comparable in achieving the monitoring goal. Among them, wireless sensors are easier to be installed (compact and lighter devices), with no need for cables for power supply and data transmission and a lot cheaper than wired sensors.

**Table 3.** Comparison between main features of wired velocimeters and wireless sensor nodes.

| Feature | Wired Velocimeters | Wireless Sensor Nodes |
|---|---|---|
| Number of axes | 2 | 2 |
| Maximum tilt | 10° vertical, 2° horizontal | Compensated through static acceleration measurements |
| Dimensions | 180 × 170 × 90 mm | 91 × 70 × 38 mm |
| Weight | 1500 g | 240 g |
| Power supply source | 12 V | Energy harvesting |
| Data transmission | Wired | Wireless |
| Cost | 350 € | Approximately 200 € |

During the months of the field test on the Bressana bridge, the sensor nodes did not exhibit any communication issue. The energy harvesting system was able to recharge the Li-Po battery when the sun was present, allowing a very long autonomy. An example of the functioning of the PV panel in recharging the battery during a typical winter day is represented in Figure 7. The graph represents data related to the transmission phase of the acquired time histories, which is the most expensive one in terms of energy consumption. In the time period between approximately 6 am and 3 pm, the PV panel mounted on the sensor node was hit by the sunlight, producing an inflow energy characterized by a positive input voltage on the board. If the focus is switched to the current incoming to the battery (where the minus sign means a recharging current), it is possible to notice that the energy

harvesting system is able, in the worst possible scenario of consumption for the board, to balance the consumption and recharge the battery with a current value up to 40 mA. Moreover, it can be noticed that the mean current flowing in the battery in this worst-case scenario is negative considering the whole day, meaning that overall recharging current is always higher than the consumption one. This represents a very significant assessment of the energy harvesting system in providing a long autonomy to the wireless sensor nodes for continuous monitoring activities.

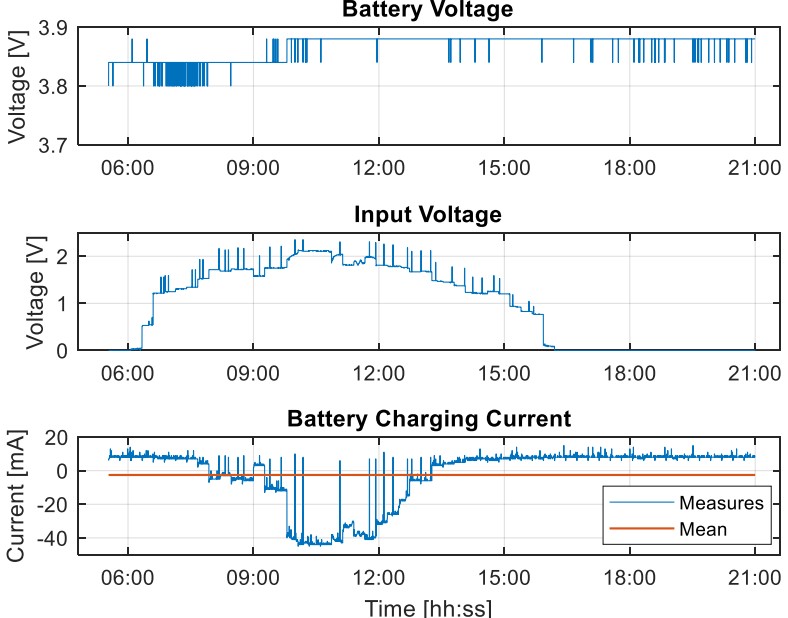

**Figure 7.** Exemplary acquired data representing the energy harvesting system charging the sensor node battery during one day (23 February 2023).

The wireless devices collected during the field test on the Bressana bridge showed several acceleration time histories triggered both by the train passages on the railway deck and probably by some passages of heavy vehicles on the upper deck. An example of these is reported, respectively, in Figures 8–10. The time history represented in Figure 8, representing the bridge response to the train passage, is characterized by higher maximum acceleration values with respect to the ones in Figure 10 (related to the traffic excitation) and by a trend from which the structural damping can be readily appreciated. Figure 9 represents a part of the structure-free response contained in the time history of Figure 8. The main excited bridge natural frequency in that case (approximately 9 Hz) can be easily identified. The time history acquisition triggered by the road traffic excitation (Figure 10) is characterized by lower maximum acceleration values, in the order of one tenth of the previous ones. This last result, obtained thanks to the adoption of the very low noise MEMS accelerometer, is significant since it allows the use of the developed system also on conventional highway viaducts (where the bridge excitation is represented only by road traffic) for modal identification purposes. The acquired time histories were then processed offline as explained in Section 4.

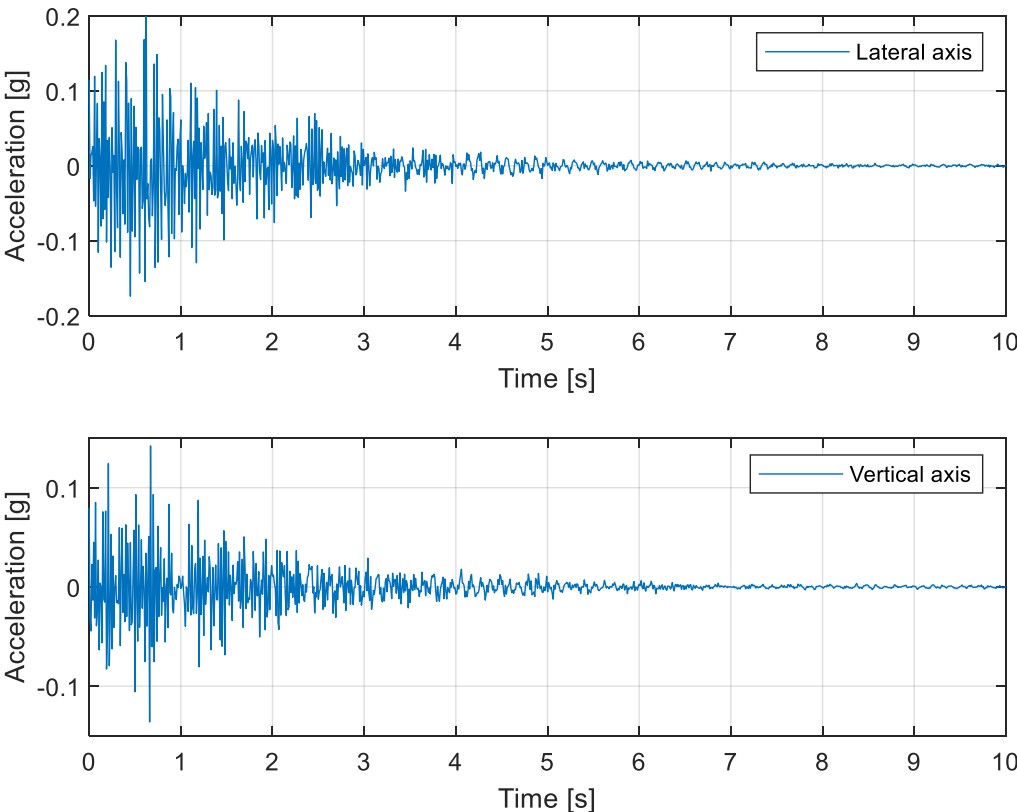

**Figure 8.** Exemplary acceleration time histories acquired by the wireless sensors triggered by the train passage on the lower deck.

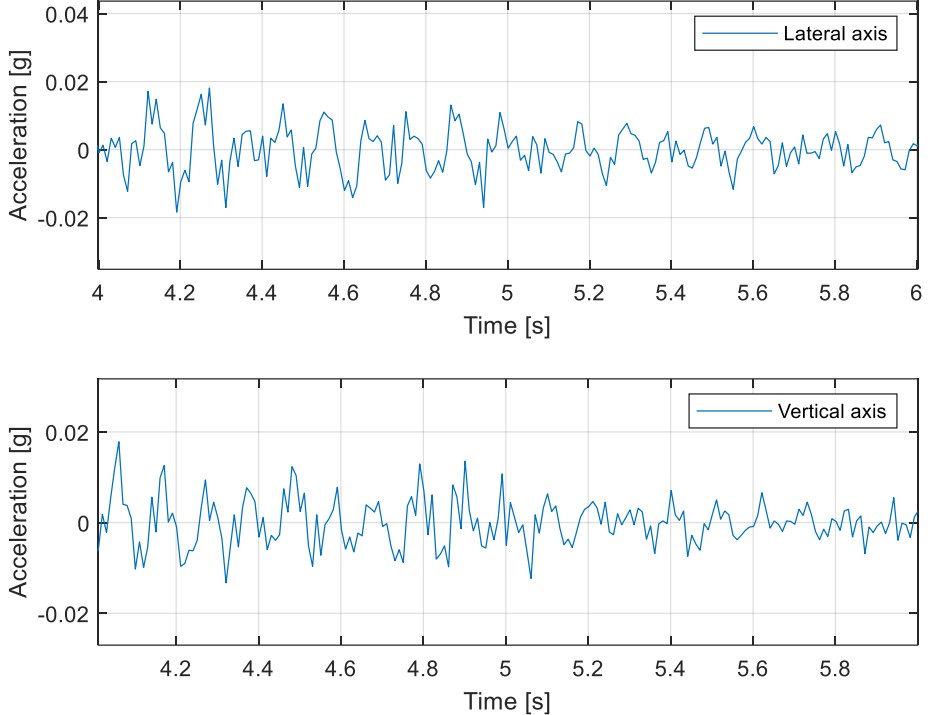

**Figure 9.** Exemplary acceleration time histories acquired by the wireless sensors triggered by the train passage on the lower deck. Focus on the free response of the structure after the excitation.

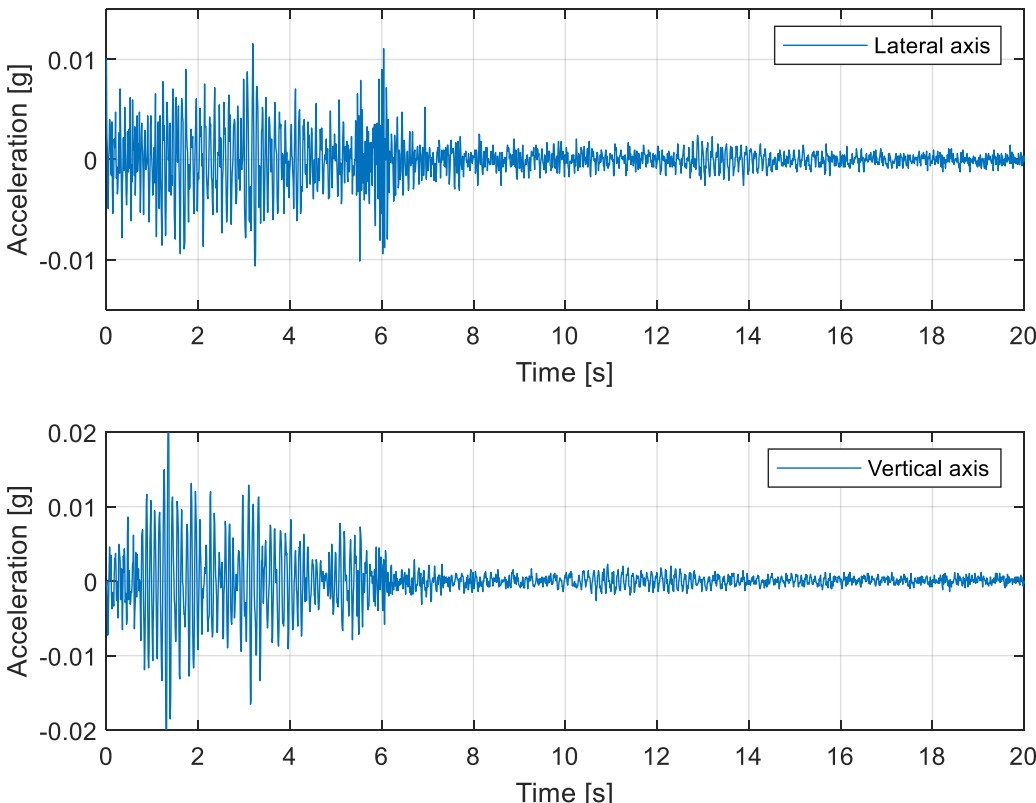

**Figure 10.** Exemplary acceleration time histories acquired by the wireless sensors possibly triggered by a heavy vehicle transiting on the upper deck.

## 4. Wireless Sensor Node Measurement Performance Assessment

The main scope of the field test was to verify if acceleration data acquired by sensor nodes were suitable to be used for modal identification of a typical railway bridge. The identification of modal characteristics is significant for SHM techniques since a change of these structural features in time can be attributed to the presence of an extensive damage in the structure [10]. In this framework, frequency domain algorithms have been the most popular, thanks to their simplicity and processing speed [39].

The goal of this analysis is to understand if acceleration data collected are significant and can be used to compute the main bridge natural frequencies. One of the simplest methods for the identification of modal parameters found in the literature is the basic frequency domain (BFD) method. This approach is also known as peak-picking method since modes are identified by picking the peaks in power spectral density (PSD) plots [40]. Since usually not all the natural frequencies are represented in an averaged PSD of a single measurement location, it is necessary to consider all measured locations to ensure the identification of the structural natural frequencies of interest. An interesting method to avoid the analysis of several PSD plots through the introduction of a new frequency indicator function based on a group of PSDs is explained in [41].

This newly defined natural frequency indicator is the function named averaged normalized power spectral densities (*ANPD*), which can be computed through Equation (1).

$$ANSPD(f_k) = \frac{1}{l}\sum_{i=1}^{i=l} NPSD_i(f_k), \tag{1}$$

*NPSD$_i$(f$_k$)* is defined by Equation (2).

$$NSPD_i(f_k) = \frac{PSD_i(f_k)}{\sum\limits_{k=0}^{k=n} PSD_i(f_k)}, \tag{2}$$

where $f_k$ represents the discrete frequency and $n$ is the number of discrete frequencies. The peaks of the *ANPSDs*, obtained through the elaboration of the data collected by wireless sensor nodes, were then used to estimate the natural frequencies of the bridge and the results of this procedure are visible for lateral and vertical axes in Figure 11.

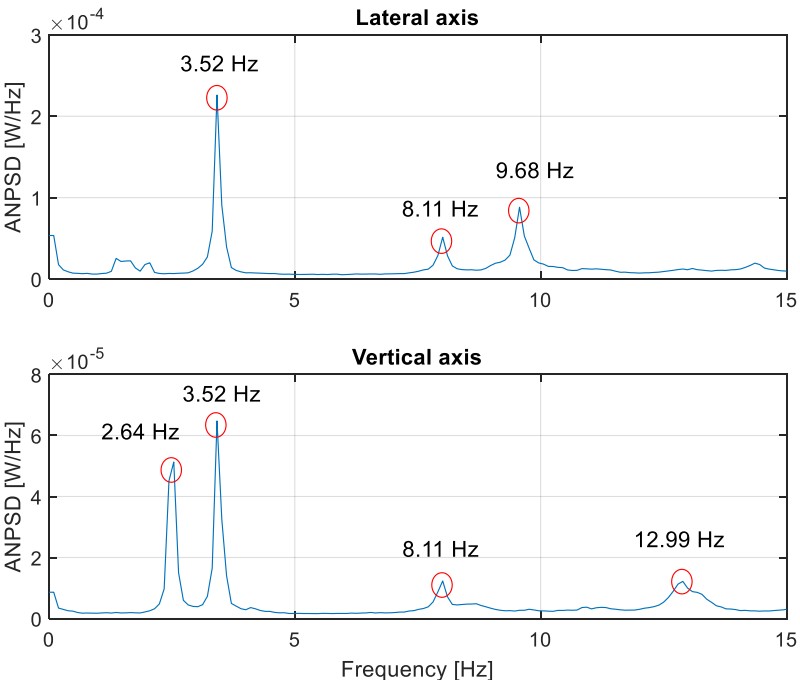

**Figure 11.** Averaged normalized power spectral densities for the lateral axis (**top**) and vertical axis (**bottom**) computed from acceleration time histories acquired by wireless sensors. Red circle represents natural frequencies identified through the peak-picking technique.

In order to assess the accuracy in the frequency estimation obtained with acceleration data collected by the wireless sensors, reference values of natural frequencies of the different modes were obtained thanks to the processing of data acquired by the wired system. The time interval in which the signals used to estimate the reference frequencies were acquired corresponds to the nighttime hours, during which there are no train passages. Thus, the input to the system was solely natural excitation from the environment and, to a very limited if non-zero extent, from road traffic. For this application, road traffic was considered a source of random noise as well.

Then, these signals were filtered and cleansed to remove anomalies, and PSDs were derived for each acquisition channel. These PSDs, coming from relatively long signals (in the order of hours) had a clear enough spectral content to estimate the natural frequencies, corresponding to the frequency value at the resonance peaks.

In particular, regarding the vertical modes, the natural frequencies of the first three bending and torsional modes are successfully estimated. For the lateral modes instead, the natural frequencies of the first three bending modes are estimated. The results are summarized in Table 4 where the second column reports the reference values obtained by the conventional wired system, while in the third column, estimations obtained from data collected by wireless sensor nodes are reported. The reference values were computed, as explained previously, by taking advantage of a large database obtained with the permanent installation of the conventional wired system on the structure for several months.

**Table 4.** Comparison between natural frequencies estimated by the wired and wireless monitoring systems.

| Vibration Mode | Wired System Estimation | Wireless System Estimation |
|---|---|---|
| 1st Vertical (bending) | 2.5 Hz | 2.64 Hz |
| 1st Lateral | 3.5 Hz | 3.52 Hz |
| 1st Vertical (torsional) | 3.5 Hz | 3.52 Hz |
| 2nd Vertical (bending) | 5.8 Hz | - |
| 2nd Lateral | 6.7 Hz | - |
| 3rd Vertical (bending) | 8.5 Hz | 8.11 Hz |
| 2nd Vertical (torsional) | 9.5 Hz | - |
| 3rd Lateral | 9.7 Hz | 9.68 Hz |
| 3rd Vertical (torsional) | 13.1 Hz | 12.99 Hz |

As can be observed from Table 2, experimental values obtained from wireless sensors are in good agreement with the reference ones obtained through the processing of signals coming from the conventional system. The fact that the frequencies related to both vertical and lateral second mode have not been detected is due to the positioning of the only two installed prototypes, namely the midspan, which represents a nodal point for second modes (Figure 4). This comparison allows the validation of the developed solution for railway bridge modal identification purposes.

As already explained, the main goal of this data analysis was the identification of the bridge natural frequencies to assess the validity of data acquired by sensor nodes through a comparison with reference ones. Once identified the natural frequencies, their values can also be monitored in time. In fact, one consequence of the presence of a structural damage is a drop in the bridge natural frequencies. An increase in the damage severity leads to a higher drop in the natural frequency [42]. Anyway, a slight decrease in the values can be associated to the variation in environmental conditions (i.e., temperature variation, etc.). Therefore, usually a damage can be detected when a variation of frequencies larger than 5% is observed [43].

In order to understand if the natural frequencies computed through the data coming from wireless sensors present a too high scatter to be potentially adopted as diagnostic parameters, their variation in time is considered. In particular, the trend in time of first lateral frequency values estimated by sensor nodes over an 18-day time window is considered in Figure 12. As can be noticed, the variation of this parameter does not present a clear decreasing trend, and it is anyway well below a ±5% variation with respect to the starting detected frequency value.

The results obtained allow the assessment of the measurement performance of the developed sensor nodes for modal identification and continuous monitoring purposes. Since the variation in time of natural frequency values represents one of the simplest SHM technique to identify the possible presence of structural issues, the FFT computation could be implemented directly on on-board sensors to send a synthetic and significant indicator of the bridge health. This on-board operation could avoid the transmission and post processing of time histories, limiting the data amount to be treated. More complex SHM algorithms will be studied in the future, and the possibility of implementing them at the gateway or sensor node level will be analyzed.

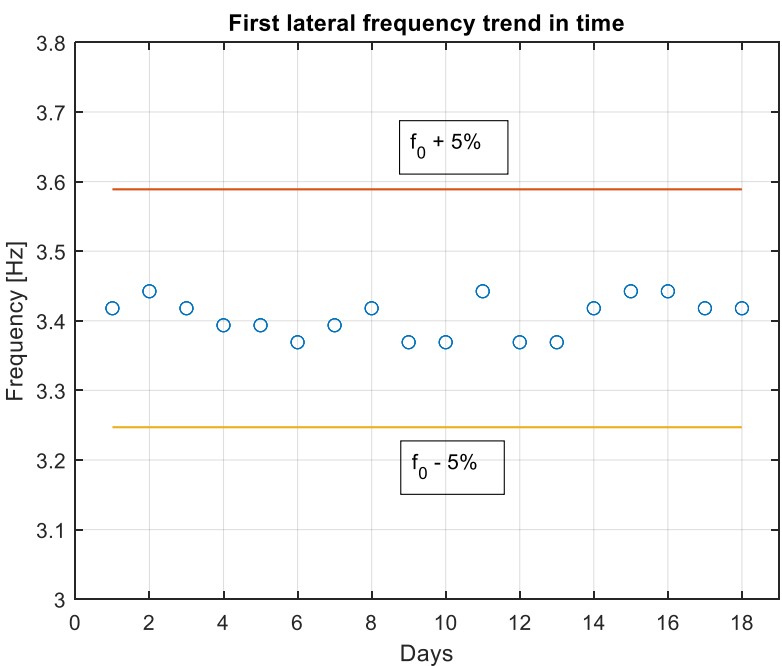

**Figure 12.** Trend of the first lateral mode natural frequency in time. The slight difference in the identified values allow the use of the variation of this parameter in time as a diagnostic indicator.

## 5. Conclusions

A smart wireless monitoring system was developed to perform long-term continuous monitoring activities on a railway bridge with modal identification aim. The system is based on smart wireless sensor nodes, relying on energy harvesting techniques for battery recharge, able to be easily mounted on the structure to perform accelerometric measurements. Another advantage of the system is the possibility to perform on-board data processing, with the aim of sending only significant information remotely. The system was tested in an experimental campaign on a railway bridge during which significant data were acquired. The analysis of data coming from the field test proved the efficiency of the energy harvesting solution in recharging the battery with continuity in time. Moreover, the measurement performance of the developed sensor nodes was demonstrated by analyzing in the frequency domain the collected time histories and by comparing the estimated main bridge natural frequencies with reference ones. This way, the developed sensor nodes could be used in place of wired sensors to perform monitoring activities using simple health indicators, such as the variation of natural frequencies in time. This would allow instrumenting a high number of bridges since wireless sensors are low cost, easy to install, not affected by maintenance during time, and endowed with a very long autonomy, thanks to the implementation of energy harvesting techniques. The further development of this technology could allow the adoption of more complex SHM techniques in the gateway present on the structure or directly on the sensor node.

**Author Contributions:** Conceptualization, F.Z., N.D. and M.B.; methodology, F.Z., N.D. and A.A.; software, F.Z., N.D. and M.M.; validation, F.Z. and N.D.; formal analysis, F.Z. and N.D.; investigation, F.Z., N.D. and M.M.; resources, M.M. and A.A.; data curation, F.Z., N.D. and A.A.; writing—original draft preparation, F.Z.; writing—review and editing, F.Z. and M.M.; visualization, F.Z. and N.D.; supervision, M.M. and M.B.; project administration, M.B.; funding acquisition, M.B. All authors have read and agreed to the published version of the manuscript.

**Funding:** This research received no external funding.

**Institutional Review Board Statement:** Not applicable.

**Informed Consent Statement:** Not applicable.

**Data Availability Statement:** Not applicable.

**Conflicts of Interest:** The authors declare no conflict of interest.

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
