# Peer review of "Development and Field Validation of Wireless Sensors for Railway Bridge Modal Identification"

_applsci, doi:10.3390/app13063620_

Round 1

Reviewer 1 Report

The article might be enhanced by a more detailed comparison between the newly proposed sensors and available analogs.

An in-depth discussion on the natural-frequency estimation procedure would be beneficial for readers who might be no familiar with the work of A. J. Felber.

 I didn’t mention an explanation for the abbreviation FFT (line 317). Please check and, if the explanation is provided, ignore this comment.

Reviewer 2 Report

A smart wireless monitoring system developed for bridges SHM is presented in this paper. But there is no novelty. It can not be accepted as the current style.

 1. The title does not summarize the whole content. After reading the whole manuscript, the main content is not the wireless monitoring system. It should be the monitoring results of a railway bridge through the system.  

2. The topic is relevant in the Structural Health Monitoring field. There is no novelty.  

3. The references are appropriate and the conclusions are consistent with the evidence and the arguments. But the content is not abundant.  

4. It is just like a test report. It is the introduction of the monitoring system in the first six pages. Then, the authors introduced the collected acceleration results. Therefore, the collected data were shown in figure six. The test data is too lacking to get much more useful results.  

5. In section 4, data analysis and discussion, the data should be analyzed and the authors should give a much more deep analysis. Now the authors just analyzed the Power Spectral Densities. The analysis is too simple. How did the author get the data from the second column in Table 2?   

6. I think the authors must measure many more data from the bridge. How do you use the data? And what are the functions of the data and your analysis results? And what are the advantages and disadvantages of the wireless monitoring system? If we use the traditional system, we can also get the same results.

Reviewer 3 Report

The paper present wireless sensors-based vibration analysis of a railway bridge 2 for Structural Health Monitoring. However this subject is not new, there are many publications relate wireless sensors for SHM. 

Lynch, Jerome P., and Kenneth J. Loh. "A summary review of wireless sensors and sensor networks for structural health monitoring." Shock and vibration digest 38, no. 2 (2006): 91-130.
Therefore, i regret to reject this paper.

Reviewer 4 Report

Manuscript ID: applsci-2244266
Manuscript Title: Wireless sensors-based vibration analysis of a railway bridge for Structural Health Monitoring purposes
General comment:
The article subject is very interesting for Structural Health Monitoring (SHM). The study was well planned, and the findings were presented and discussed correctly. It adds valuable remarks for the area of wireless sensors-based SHM. Therefore, it can be accepted for publication after carrying on some revisions. Here are some recommendations for improvement.
Specific comments:
1. The abstract must reflect the fundamentals stated in the paper itself. The discussion could be further described, the main findings should be introduced.
2. It is necessary to add a methodology flowchart describing the research in detail.
3. The analysis of the test data was not detailed enough, and the reasons affecting the test results were not discussed in depth.
4. The conclusion does not provide any visible scientific contribution. Much stronger conclusions must be drawn from the research results presented.

Round 2

Reviewer 2 Report

The authors has answered all the questions. It can be accepted.

Author Response

Thanks for your contribution in the review process.

Reviewer 3 Report

1.     Please clarity more contribution of this paper?

2.     In the section 3, please show clearly the results of experience for the Bressana railway bridge?

3.     Please provide more the actual pictures for this test?

4.     Make clearly the data base that show in the table no.2 and where did you get them from?

5.     How to verify the results of your paper?

6. The introduction shoud be revised. There are some papers related SHM problems such as: https://doi.org/10.1016/j.advengsoft.2022.103363; https://doi.org/10.1016/j.engfailanal.2022.106829

Reviewer 4 Report

The authors considered the comments of the reviewer. The revised manuscript is improved. I have no further comments.

Author Response

(The authors gave the same response as above.)
